# Sestrins in Carcinogenesis—The Firefighters That Sometimes Stoke the Fire

**DOI:** 10.3390/cancers17091578

**Published:** 2025-05-06

**Authors:** Alexander Haidurov, Andrei V. Budanov

**Affiliations:** School of Biochemistry and Immunology, Trinity Biomedical Sciences Institute, Trinity College Dublin, Pearse Street, D02 R590 Dublin, Ireland

**Keywords:** Sestrins, mTOR, cancer

## Abstract

Sestrins (SESN1-3) are a family of stress-responsive proteins that serve as direct targets of several transcription factors, including tumour suppressor protein p53. Structural studies have identified three core molecular functions of Sestrins: scavenging of reactive oxygen species (ROS), inhibition of mTORC1 signalling, and leucine binding. As leucine-dependent mTORC1 inhibitors and antioxidants, Sestrins regulate metabolism and redox balance, both of which are frequently dysregulated in cancer. Dysregulation of Sestrin expression, most commonly downregulation, is widely observed in cancers and is often associated with increased tumour growth and poor prognosis. However, in certain contexts, Sestrin overexpression may promote tumour survival, and this review explores the molecular mechanisms underlying these seemingly contradictory effects on carcinogenesis, evaluates the evidence for Sestrin involvement across various cancer types, and discusses the potential of Sestrins as diagnostic markers and therapeutic targets.

## 1. Introduction

Sestrins are a family of stress-responsive proteins discovered in the 1990s. There are three Sestrin genes (*SESN1-3*), which encode functionally similar proteins but are regulated by different transcription factors and environmental stimuli. Sestrins are ubiquitously expressed among most human tissues and the early Sestrins can be traced back to invertebrate animals, such as *Caenorhabditis elegans* [1]. With the arrival of vertebrates, the single Sestrin gene appears to have duplicated into the three Sestrin genes we know today, presumably to provide redundancy and broaden the range of responses, as each Sestrin gene is regulated by a distinct set of transcription factors [2].

Sestrins are built from two somewhat symmetrical domains, which serve distinct, yet concordant molecular functions (Figure 1). The three established molecular functions of a Sestrin protein are direct reactive oxygen species (ROS) scavenging, sensing of cellular leucine, and inhibition of the mammalian target of rapamycin complex 1 (mTORC1). Sestrins directly neutralise hydrophobic ROS molecules [3,4], but may also support antioxidative cell signalling. They also directly bind leucine via their leucine-binding pocket and thus change their conformation [5]. The conformational change prevents their inhibition of mTORC1, as they are unable to associate with their partner—the GATOR2 protein complex, through which they inhibit mTORC1 [6]. Their inhibition of the mTORC1 complex pauses catabolic metabolism and protein synthesis, redirecting the cell from growth to recycling and repair, thereby halting proliferation and biomass accumulation while promoting cellular restoration through the endogenous recycling process—autophagy [7].

mTORC1 is tightly regulated by two sets of small GTPases: RHEB and RagA-D, through separate signalling pathways. RHEB, anchored to the lysosomal membrane, becomes active in response to ATP, cytokines, glucose, growth factors, and endocrine signals. These signals regulate RHEB through the tuberous sclerosis complex (TSC), which consists of TSC1, TSC2, and TDC1D7. TSC functions as a GTPase-activating protein (GAP) for RHEB, converting active GTP-bound RHEB to its inactive GDP-bound state [8,9]. Growth factor signalling activates the AKT kinase, which phosphorylates TSC2, reducing TSC activity and relieving its inhibition of RHEB and mTORC1 [10,11,12]. In contrast, the AMP-activated protein kinase (AMPK) complex, a central sensor of the AMP/ATP ratio, inhibits mTORC1 indirectly by activating TSC2 and phosphorylating the mTORC1 subunit RAPTOR [13,14,15]. Thus, AKT, and AMPK act in opposition. Early studies indicated that SESN2 may inhibit mTORC1 via AMPK. SESN2 reduces RHEB activation in a TSC2-dependent manner and interacts with both TSC and AMPK, indicating its association with these complexes. A mutant form of SESN2 lacking the N-terminal fails to bind AMPK [16]. SESN2 may facilitate AMPK activation by acting as a platform for LKB1-mediated phosphorylation of the AMPK catalytic subunit [17]. However, the precise SESN–AMPK mechanism remains unclear.

In addition to intercellular signals, mTORC1 is also regulated by an intracellular amino acid-sensing axis. The Rag GTPases function as heterodimers, with RagA/B bound to GTP and RagC/D bound to GDP in their active state [18]. The Ragulator complex anchors Rags to the lysosome [19]. The protruding RAPTOR subunit of mTORC1 binds to the active Rags complex, docking mTORC1 at the lysosome, which is essential for its activation by lysosome-anchored RHEB [20]. The activity of Rags is controlled by the GATOR complexes (GATOR1 and GATOR2). In the absence of leucine, Sestrins bind GATOR2, lifting its inhibition of GATOR1. GATOR1 functions as a GTPase-activating protein (GAP) for RagA/B, inactivating it and preventing mTORC1 activation [6,21,22,23,24].

In addition to mTORC1, the mTOR kinase also forms a second distinct complex called mTORC2. While both complexes house the same mTOR protein in mammals, the accessory subunits of mTORC2 are distinct from those of mTORC1. Instead of RAPTOR, which binds mTORC1 to Rags at the lysosomal membrane, mTORC2 contains RICTOR and thus is not recruited to the lysosome. The precise subcellular localisation of mTORC2 remains unclear, although evidence suggests that its mSIN1 subunit may direct the complex to the plasma membrane [25], and it has additionally been shown to localise at the endoplasmic reticulum [26]. mTORC2 has a distinct set of substrates. Through PKC, this complex influences cytoskeletal organisation [27,28], thus affecting the metastatic potential and survival of some cancers [29]. Notably, mTORC2 also phosphorylates AKT, a protein whose activity is frequently promoted by Sestrins in cancers [30,31]. While initial findings demonstrated that silencing RICTOR dramatically diminished AKT activation in response to SESN2 induction in the MCF7 breast carcinoma cell line [32], the precise role of mTORC2 in Sestrin-mediated AKT regulation continues to be explored [33,34].

While Sestrins contain a catalytic cysteine which is essential for neutralisation of hydrophobic ROS, such as cumene hydroperoxide [3,4], they also may support other antioxidative mechanisms. Sestrins were shown to support the activation of the antioxidant transcription factor Nrf2 [35,36]. Some experiments indicated that SESN2 may interact with Keap1, the inhibitor of Nrf2, and induce Keap1’s selective degradation, thereby activating Nrf2 [35,37]. Other studies supported the notion that Sestrins protect mitochondria from ROS-induced damage. Mitophagy, a selective form of autophagy that recycles damaged mitochondria to maintain cellular function, has been proposed to involve Sestrins. One study showed that Sestrins promote mitophagy of impaired mitochondria in macrophages [38], and another work has demonstrated that Sestrins colocalise with mitochondria [39]. Also, SESN2 has been shown to directly support mitophagy, through association with ULK1, which promotes Beclin-1 phosphorylation, thereby initiating mitophagy via the PINK1–Parkin pathway [40].

While Sestrins are expressed at varying basal levels in most tissues [41], they are drastically upregulated in response to a multitude of stress insults (Figure 2). Most important for regulation of carcinogenesis are the p53-inducible *SESN1* and *SESN2* genes, which are direct targets of p53 [1,42] and are both expressed at low levels in p53-deficient cells [1,43]. The *TP53* gene is frequently mutated in cancer, with over 29,000 identified mutations and an overall prevalence of 29% across all cancers [44]. While p53 mutations vary by cancer type, they are nearly ubiquitous in malignancies, with rates varying between 10% in haematological malignancies to over 90% in high-grade serous ovarian carcinoma [45]. As oxidative damage can inadvertently damage DNA [46], *SESN1* and *SESN2* have been found to be essential for the protective effect against oxidative damage mediated by p53 [43]. Hypoxia, under which tumours frequently find themselves, steadily upregulates *SESN2* [1]. In some cases, *SESN2* expression has been shown to correlate with HIF-1α activity; however, a direct regulatory relationship remains unconfirmed [47,48]. ER stress upregulates *SESN2* through transcription factors ATF6 [49], XBP1 [50], and ATF4, for which a *SESN2* promoter region has been identified [51]. Oxidative stress can upregulate *SESN2* via the Nrf2 antioxidative transcription factor [52], and the *SESN3* gene is a confirmed direct target of the antioxidative transcription factors FOXO1&3 [53,54].

Recent statistics have shown that approximately 20% of people in the world will develop cancer in their lifetime [55], underscoring the urgent need for reliable diagnostic markers and therapeutic targets. The multifaceted nature of Sestrin proteins and the network of signals governing their upregulation complicate their physiological and pathological roles. In normal cells, their functions reinforce each other, as their antioxidative activity protects cells from damage while mTORC1 inhibition halts catabolic processes, thereby shifting the cell into a protective recovery state. In cancer cells, Sestrins hamper aberrant growth, promote DNA-surveillance mechanisms by facilitating apoptotic cell death, and limit oxidative stress to protect genomic integrity. However, when Sestrin activators such as p53 are nonfunctional, their tumour-suppressive capacity is compromised. Reduced Sestrin expression, which is frequently observed across many tumours, impairs their ability to inhibit mTORC1 effectively. In other cases, disruptions in their signalling partners such as GATOR1 may prevent effective mTORC1 inhibition. As a result, when Sestrins fail to inhibit mTORC1, they may be repurposed by the tumour to support its growth and survival, with their ability to protect against oxidative stress and metabolic challenges inadvertently promoting tumour progression.

Overall, we observe that Sestrins are downregulated across multiple cancers. According to the database GENT2, the three *SESN* genes are downregulated across all cancers [56]. Poor expression of Sestrins is linked to poor prognosis in several cancers such as lung and colorectal cancers [30,57,58]. Yet, conversely, we encounter cases in which raised Sestrin expression can carry a poor prognosis [31,59]. For example, in melanoma, Sestrin can support the survival of cancerous cells exposed to ultraviolet B (UVB) radiation, and also protect metastatic cells from oxidative damage, thus supporting growth and survival of the cancer [31,60].

This review will explore the available evidence and attempt to clarify the seemingly contradictory roles of Sestrins in cancer, highlighting the molecular mechanisms that underlie their dual function as both tumour suppressors and facilitators of malignancy.

## 2. Sestrins in Cancer

Carcinogenesis is distinguished by critical hallmarks such as continuous cell growth, resistance to apoptosis, evasion of tumour suppressors, genomic instability, and persistent inflammation, among others [61]. Sestrins act as crucial mediators of the p53 pathway, influencing these key hallmarks of carcinogenesis [62,63]. However, their influence on cancer can be ambiguous depending on the context. Sestrins inhibit mTORC1 while also stimulating the AKT pathway [63], both of which are frequently activated in cancers and can have opposing effects on tumorigenesis, either supporting or suppressing it. These parallel but interdependent pathways indicate that Sestrins could inhibit tumour growth by repressing mTORC1 or promote survival and proliferation through AKT activation [64]. Additionally, their role in autophagy activation can either deter or facilitate carcinogenesis, depending on the carcinogenesis stage, and type [65]. Consequently, the impact of Sestrins on cell viability may differ based on the stress type, resulting in varying effects within tumours [66,67].

### 2.1. Sestrins in Follicular Lymphomas

The 6q21 locus, which includes *SESN1*, frequently undergoes deletion in human follicular lymphomas, and *SESN1* expression is notably reduced in lymphomas that express the mutant EZH2X641Y protein [68]. EZH2, a crucial component of the Polycomb Repressive Complex 2 and a histone-lysine N-methyltransferase, catalyses the methylation of histone H3 on lysine 27, effectively repressing gene expression—a process vital for lymphocyte maturation [69]. The EZH2X641Y mutation in lymphoma cells leads to mTORC1 activation via suppression of *SESN1* transcription. This activation of mTORC1 is implicated in the progression of cancer, as evidenced by the observation that mTORC1 inhibition suppresses stem cell proliferation and lymphoma advancement [70]. Furthermore, the knockout of *SESN1* in the SU-DHL-10 lymphoma cell line abolished the apoptosis induced by GSK126, an inhibitor of EZH2’s methyltransferase activity [68].

### 2.2. Sestrins in Lung Cancer

SESN1 and SESN2 frequently exhibit reduced expression in lung tumours, with decreased SESN2 levels marking a poor prognosis for lung cancer patients [30,58]. The loss of SESN1 and SESN2 in A549 lung adenocarcinoma cells facilitates tumour growth and enhances resistance to apoptosis [30,43,66]. Interestingly, *Sesn2* knockout in mice suppresses early-stage lung tumour growth, which may be linked to diminished Akt kinase activity [30]. However, in the A549 cell line obtained from advanced lung adenocarcinoma, SESN2 did not contribute to the activation of AKT and sensitised cancer cells to cell death and pro-apoptotic cytokines [30,43,66]. A recent study in A549 cells suggests that SESN1 and SESN2 regulate aberrant STAT3 activation in lung cancer, limiting malignant proliferation and enhancing sensitivity to apoptosis [71].

### 2.3. Sestrins in Liver Cancer

Unpublished findings from Budanov and Karin suggested that *Sesn2* knockout in mice leads to reduced liver tumour growth in diethylnitrosamine (DEN)-induced liver carcinogenesis models. In contrast, a later study indicated that increased expression of *Sesn3*, another member of the Sestrin family, correlates with improved survival rates in animal models of liver cancer [72]. Experiments involving mice injected with the carcinogen DEN and subsequently fed a choline-deficient high-fat diet showed that *Sesn3* knockout animals developed a greater number of tumours than control subjects, and these tumours were also larger and more metastatic, underscoring the tumour-suppressive role of the *Sesn3* gene [72].

### 2.4. Sestrins in Prostate Cancer

While SESN3 exhibits antiproliferative properties, it has been identified as upregulated in hormone-refractory prostate cancers [73]. This increase in *SESN3* expression occurs in prostate carcinoma cells transitioning into hormone-refractory neuroendocrine prostate cancer (NEPC). Such changes may be associated with the reduced expression of micro-RNA-708, which directly interacts with *SESN3* mRNA, thereby diminishing its expression [74]. NEPC is a highly aggressive form marked by loss of androgen signalling, gain of neuroendocrine phenotype, and resistance to hormone therapies [75]. *TP53* loss is a common feature of NEPC, found in about 67% of cases [76], and likely leads to a reduction in *SESN1* and *SESN2* expression. In this context, the p53-independent *SESN3* may be upregulated to manage oxidative stress. Since this occurs during the transition to NEPC, mTORC1 inhibition by Sestrins may no longer be a concern, making SESN3’s antioxidant role selectively advantageous.

### 2.5. Sestrins in Skin Cancer

In contrast to Sestrins’ recognised tumour-suppressive function across multiple tissues, SESN2 frequently shows elevated expression in various skin cancers, including melanomas and squamous cell carcinomas, acting as an oncogene to promote skin carcinogenesis [31]. This effect may be attributed to SESN2’s protective role against ultraviolet and other stressors that threaten cancer cell viability, primarily through the activation of AKT [31,60]. In sebaceous carcinoma, SESN2 was found to be significantly downregulated as a result of p53 mutations, and low SESN2 expression correlated with poor tumour differentiation, which is an indicator of tumour aggressiveness and poor prognosis [77].

### 2.6. Sestrins in Colon Cancer

Expression analysis of SESN2 in colorectal cancers revealed its downregulation, with its reduced levels linked to advanced tumour stages marked by vascular invasion, lymph node metastasis, and liver metastasis. Furthermore, lower SESN2 expression correlates with a poor prognosis and a reduced patient life expectancy [57]. Since this downregulation was observed in advanced stages of cancer, it is likely attributable to p53 inactivation, which is frequently observed in later stages of tumour progression. Another study showed that *Sesn2* knockout in a mouse model where colorectal cancer was induced through dietary administration of dextran sodium sulphate (DSS) led to accelerated tumour growth. This growth was associated with increased mTORC1 activity and inflammation in the colon [78].

### 2.7. Sestrins in Endometrial Cancer

A study of SESN2 in primary endometrial cancer tissue revealed that SESN2 was upregulated in cancer tissues. Interestingly, mTORC1 activity remained elevated in tumour samples despite increased SESN2 expression. Subsequent experiments on endometrial cancer cell lines (HEC-1-A and Ishikawa) produced contradictory results: silencing SESN2 in these cell lines enhanced mTORC1 activity, increased proliferation, and raised intracellular ROS levels. Furthermore, SESN2 silencing promoted the growth of larger and heavier xenografts from these cell lines. Although SESN2 functions as expected in endometrial cell lines, SESN–mTORC1 signalling may be disrupted in the tumour microenvironment, as SESN2 levels do not correlate with reduced mTORC1 activity in endometrial tumours [79]. Another study has shown that *SESN3* is highly methylated in endometrial cancers, suggesting that its epigenetic repression may facilitate oncogenic processes in this context [80].

### 2.8. Sestrins in Brain Cancer

In the context of neuroblastoma, a study showed that lysine-specific demethylase 1 (LSD1) influences the histone modifications of the *SESN2* promoter, thereby repressing its expression. Inhibition of LSD1 was shown to activate autophagy and repress mTORC1, and SESN2 was required for this effect. Furthermore, the study found an inverse correlation between LSD1 and SESN2 expression in differentiated neuroblastomas, and reported that high LSD1 and low SESN2 levels were each significantly associated with poorer survival [81]. In glioblastoma, silencing SESN2 in U87 tumour cells sensitised them to ionising radiation (IR), reducing proliferation, and significantly increasing ROS levels. SESN2 silencing also elevated platelet-derived growth factor receptor beta (PDGFRβ) expression. Mechanistically, SESN2 was shown to promote proteasomal degradation of PDGFRβ, as its ubiquitination and degradation were impaired upon *SESN2* knockdown. Co-silencing SESN2 and PDGFRβ partially normalised ROS levels and proliferation, although not to baseline [82]. These findings are supported by studies in mouse lung fibroblasts, where *SESN2* knockout increased PDGFRβ expression and this increase was reversed by SESN2 reconstitution. Notably, reconstitution with the catalytically inactive SESN2 mutant (C125G) that is incapable of reducing ROS did not affect PDGFRβ expression, indicating SESN’s ROS regulation as the potential mechanism [83]. Further work demonstrated that *SESN2* knockout impaired PDGFRβ degradation via reduced proteasomal activity, distinct from the mechanism observed in glioblastoma. Moreover, SESN2 may modulate PDGFRβ expression indirectly through Nrf2 activation and antioxidant gene induction [36]. Overall, the interaction between SESN2 and PDGFRβ in glioblastoma remains incompletely understood, and it is unclear whether PDGFRβ modulation by SESN2 significantly influences glioblastoma progression or reflects a broader regulatory mechanism.

### 2.9. Sestrins and Cancer Stem Cells

Tumours are recognised for their heterogeneous cell composition, where a single cell can be markedly genetically different from the rest of the tumour. Various cell types collaborate to support the tumour with adaptive mechanisms for survival and propagation within the host. Recent research highlights that cancer growth and spread are often propelled by cancer stem cells. They are noted for their unlimited proliferative capacity, stress resistance, and ability to differentiate into various cancer cell subtypes. These cells also exhibit heightened resistance to cancer treatments [84]. Putative cancer stem cells, or so-called tumour-initiating cells with stem-like properties, have been identified in a range of cancers, including leukaemia [85], lung carcinoma [86,87], hepatocarcinoma [88,89], colon carcinoma [90], melanoma [91], squamous carcinoma [92], prostate carcinoma, and other types of cancer [84]. While the involvement of cancer stem cells in follicular lymphoma remains debated, tumour-initiating cells possessing stem-like qualities have been reported for these cancer types [93].

The role of Sestrins in the regulation of cancer stem cells remains unclear. However, a study examining the activity of Sesn3 in liver cancer investigated its impact on cancer stemness. Tumours lacking *Sesn3* exhibited increased activation of Akt and Stat3, along with elevated levels of stem cell markers such as Acta2, Cd44, and Cd133 [72]. Moreover, the same study showed that SESN3 might regulate stemness by sequestering the transcription factor Gli2 in the cytoplasm through direct protein–protein interactions. However, as the SESN3–Gli2 interaction was shown via an overexpression study in 293T cells, this hypothesis requires validation using endogenous protein from liver cells [72]. Nevertheless, the support of stemness through the regulation of Gli2, AKT, and STAT3 may represent a crucial pathway through which SESN3 may counteract liver carcinogenesis. Overexpression of SESN2 in colorectal cell lines (HCT-116, SW620) lowered both mRNA and protein levels of stemness-related Sox2, Oct4, Cxcr4, and CD44. The same study showed that mouse xenografts of SESN2 overexpression had lower protein levels of Sox2 [94].

A summary of Sestrin expression changes, genetic manipulations, and biological effects across the tumour types discussed above is provided in Table 1.

### 2.10. Sestrins’ Role in Early Stages of Carcinogenesis

Despite their importance in suppressing cancer, Sestrins seem to have a limited impact on tumour initiation. A study on lung cancer in a mouse model showed that knockout of either *Sesn1*, *Sesn2*, or both does not impact tumour formation [30]. In a study of the impact of Sestrins in colon cancer, mice knockouts of *Sesn2* did not develop more tumours [78]. However, the same study revealed a crucial finding: *Sesn2* expression in the colon epithelium is essential, as its absence compromises the colon lining, leading to increased chronic inflammation and subsequent tumour formation [78]. Additionally, while Sestrins may not directly influence tumour initiation they support the integrity of DNA. Sestrins were shown to be the frontline in p53-mediated protection of DNA from oxidative damage [43], and another study showed that inhibition of Sestrins directly contributed to chromosome breaks due to the build-up of ROS in Ras-transformed fibroblasts [95]. Therefore, current evidence suggests that Sestrins do not have a direct molecular role in tumour initiation. However, they appear to play a protective role against cancer-promoting events. Thus, while Sestrins are important for maintaining tissue integrity, their role seems more indirect, as they mitigate cancer-promoting events rather than directly influencing tumour initiation.

### 2.11. Sestrins Inhibit Tumour Growth

#### 2.11.1. Sestrin—Tumour Suppressor

Although the absence of Sestrins typically does not affect tumour initiation, evidence shows that Sestrin expression has a pronounced influence on tumour volume, weight, and cell proliferation. The earliest studies on Sestrins showed that inhibition of SESN1 accelerated the proliferation of fibroblast cells [16]. Subsequent studies explored the role of Sestrins in specific cancer models. Detailed studies of inflammation-associated colon cancer showed that mice knockout of *Sesn2* did not influence tumour initiation but dramatically increased the size of the tumours [78]. Several xenograft experiments on colorectal cancer cells showed that SESN2 overexpression dramatically decreased the size and weight of tumours [94,96]. Knockout of *SESN1&2* increased the proliferation of A549 lung cancer cells, with the rate of proliferation increasing progressively with each Sestrin knocked out [30]. A subsequent study revealed that this proliferation is facilitated by altered STAT3 signalling in the absence of Sestrin [71]. Xenografts of endometrial cancer also have shown to grow larger and heavier with inhibition of SESN2 [79].

A likely molecular mechanism underlying Sestrins’ repression of aberrant growth is their inhibition of mTORC1, as this complex is frequently overactivated in cancer and essential for cell growth and biomass accumulation [97,98]. Loss of SESN2 increased mTORC1 activation and correlated with bigger tumours in the context of CRC [78]. Loss of p53, a common event in cancer progression, promotes mTORC1 activation [99]. Inhibition of overactivated mTORC1 can display tumour-suppressing effects in vivo and is therefore a common target for clinical trials [100]. In addition to inhibiting mTORC1, a recent study shows that Sestrins may also be inhibitors of STAT3 phosphorylation, thereby constraining aberrant cell proliferation [71]. Multiple studies demonstrate that Sestrins influence the cell cycle. Overexpression of SESN1/2 increased the distribution of cells in the G1 phase (growth phase) in the 293T cell line [16]. This effect could be related to expression of the cell cycle protein Cyclin D1, which Sestrins appear to suppress in particular contexts [16,71]. *Sesn* overexpression in the gut stem cells of the *D. melanogaster* also arrested cells in the G1 phase, which the authors suggested was due to mTORC1 inhibition [101]. Therefore, a substantial body of evidence supports the role of Sestrins as suppressors of tumour growth and proliferation.

Given the role of Sestrins in suppressing tumour growth and proliferation, cancers may develop mechanisms to bypass Sestrin-mediated inhibition of mTORC1 (Figure 3). One likely strategy is to decrease the activity of Sestrin-activating transcription factors, such as p53, thereby reducing the availability of Sestrins that inhibit cell growth. Another potential mechanism involves the dysfunction of GATOR1, the protein complex through which Sestrins regulate mTORC1. GATOR1 was initially identified as a tumour suppressor [22]. Moreover, the genes encoding subunits of GATOR1 have been found to be mutated and downregulated in certain cancers [102,103,104]. Although the frequency and specific effects of GATOR1 dysfunction in cancer remain to be fully explored, it is possible that an inactive GATOR1 could impair Sestrins’ ability to suppress tumour growth. Conversely, inactivation of GATOR2 in cancer could mimic Sestrin activation by lifting the inhibitory effect on GATOR1. A study demonstrated that knockout of the WDR59 subunit of GATOR2 significantly reduced the size of mammary tumours, while overexpression of SESN3 produced a similar effect [105]. Therefore, dysfunction of the SESN–GATOR–Rags signalling axis may serve as a mechanism to promote tumour growth. Post-translational modifications can influence the inhibition of mTORC1 by Sestrins. A recent study shows that K63-chain ubiquitination of SESN2 regulates its interaction with GATOR2. In colorectal cancer, this ubiquitination may be disrupted by dysregulation of the E3 ligase RNF167 and the deubiquitinase STAMBPL1, thereby impairing SESN2-mediated inhibition of mTORC1 [106]. Further research into Sestrin post-translational modifications could provide additional insights into how tumours evade Sestrin-mediated mTORC1 inhibition.

#### 2.11.2. Sestrin—Tumour Protector

Although most evidence shows that Sestrins suppress tumours, some specific cases have shown that Sestrins can instead support tumour growth. Experiments on melanoma xenografts in mice showed that cancer cells with silenced SESN2 produced smaller tumours, which the authors attributed to increased survival of inoculant cells due to SESN2’s protection from ROS and possibly the activation of pro-survival AKT [31]. Further studies of melanoma showed that Sestrins support the survival of metastatic cancer cells. The authors showed that intracellular ROS was significantly increased in suspended cells with silenced SESN2, and ROS correlated with the number of metastases induced via intravenous injection of cancer cells [59].

The antioxidative function of Sestrins may enable tumour survival by protecting aberrant cells, particularly when mTORC1 signalling is compromised. For example, a study of primary endometrial tumours showed that *SESN2* expression was correlated with lower patient survival, and mTORC1 activity levels were higher in these tumours in the presence of Sestrins, suggesting dysfunctional Sestrin–mTORC1 signalling [79]. As previously discussed, Sestrins can support Nrf2 activation and a fraction of the Sestrin proteins was found to translocate into the nucleus and possibly associate with the transcription factor Nrf2, where it had a supporting effect on its activity [36]. Overactivation of Nrf2 in cancer leads to increased expression of metabolic enzymes, which could support metabolism in favour of cell proliferation [107].

Another example of Sestrin supporting tumour growth is in lung cancer under glutamine deprivation. SESN2 was significantly upregulated in multiple glutamine-deprived lung cancer cell lines. Furthermore, *SESN2* knockouts of these lines were significantly more sensitive to glutamine deprivation-induced cell death and formed markedly smaller tumours under inhibition of the glutamine transporter [33]. This study, along with another, attributed this effect to Sestrins’ support of AKT activity [33,34], potentially via a possible association with mTORC2 via the GATOR2 complex [34], suggesting that while Sestrin–mTORC1 signalling is well defined, their cooperation with mTORC2 could be an important area for further cancer research.

Therefore, common patterns in cases where Sestrins support cancer include impaired mTORC1 signalling, activation of mTORC2 and AKT, and protection against metabolic and oxidative stress (Figure 4). Future studies on these mechanisms could clarify why Sestrins sometimes promote tumour survival and growth instead of suppressing tumour progression.

## 3. Sestrins as Therapeutic Targets in Cancer

Sestrin activity is tightly regulated by leucine sensing, and pharmacological modulation of the Sestrin–GATOR2 interaction has emerged as a potential therapeutic strategy. Small molecules such as leucine analogues have been developed to mimic or antagonise leucine’s effect on Sestrins.

Navitor Pharmaceuticals’ lead compound, NV-5138, targets the leucine-binding pocket of SESN2 and binds with comparable affinity to leucine. Structural studies revealed that SESN2 bound to NV-5138 adopts a conformation essentially identical to that of the leucine-bound protein. Unlike leucine, NV-5138 is neither incorporated into protein synthesis nor metabolised by the branched-chain amino transaminase, which is the possible reason for its high bioavailability in the brain. Thus, NV-5138 is being investigated for activation of mTORC1 in neurological disorders associated with reduced mTORC1 activity, including depression [108]. NV-5138 demonstrated antidepressant effects in rodent models in an mTORC1-dependent manner [109]. However, the utility of leucine analogues in cancer remains uncertain. Given that mTORC1 is frequently hyperactivated in malignancies, and that Sestrin-mediated mTORC1 inhibition is its core mode of tumour suppression, antagonising Sestrins may inadvertently promote tumorigenesis. Furthermore, leucine mimetics are unlikely to impact Sestrins’ antioxidative function, which may be exploited by tumours to mitigate oxidative stress.

Navitor’s patent (AU2021215177B2) describes several compounds classified as “leucine antagonists” for the SESN–GATOR2 interaction, defined as molecules that increase SESN2–GATOR2 binding by more than 40% [110]. Given that impaired SESN–GATOR2 interaction is associated with enhanced mTORC1 activity and worsened cancer progression, molecules that strengthen this interaction may have therapeutic potential in oncology. However, NV-5138 remains the company’s lead candidate and is currently in Phase 2 clinical trials. It is unclear whether the development of leucine antagonists for oncological applications is planned; nonetheless, evaluating their potential anti-cancer properties represents an intriguing research avenue.

Similarly, a cosmetic industry patent (US20170275694A1) seeks to identify compounds that upregulate Sestrin expression to combat skin ageing and pigmentation [111], although no compounds have been reported in connection to this patent.

Several natural and pharmacological agents have been identified as activators of Sestrin expression. Resveratrol, found in red grapes and wine, activates *SESN2* at concentrations of 10–30 µM [112]. However, its poor oral bioavailability (~0.3% as free unconjugated product) limits plasma concentrations to approximately 300 nM [113]. Quercetin, abundant in fruits and vegetables, induces *SESN2* expression in HCT116 cells at 50–100 µM [114,115] but similarly suffers from low bioavailability (typically ~0.5–2 µM plasma concentration) [116].

Pharmacological agents such as nelfinavir, an ER stress inducer, and bortezomib, a proteasome inhibitor, have been shown to activate *SESN2* via the ATF4 transcription factor [117]. However, therapeutic concentrations of nelfinavir (~15 µg/mL) exceed typical plasma levels achieved in patients (3–4 µg/mL) [118,119]. Bortezomib requires intravenous or subcutaneous administration, limiting its application solely as a SESN2 activator.

Metformin, a widely used anti-diabetic agent, activates *SESN2* through ATF4 induction and suppression of the SESN2 negative regulator miR-21-5p [120,121]. With high oral bioavailability and tolerable side effects, metformin remains an attractive candidate, although its efficacy as an anti-cancer adjuvant remains inconclusive [122], and its utility for *SESN2* activation in oncological contexts requires further investigation.

Conversely, in contexts where Sestrin activity supports tumour survival, strategies to downregulate Sestrins may be beneficial. For example, cabazitaxel, a microtubule-binding agent used in cancer therapy, was shown to moderately reduce *SESN3* expression, thereby increasing intracellular ROS levels [123].

In addition to direct pharmacological modulation, tumour leucine metabolism may indirectly influence Sestrin activity. Cancers can display elevated levels of leucine [124], and leucine transporters such as LAT1 (SLC7A5) are frequently found overexpressed across various cancers [125]. Elevated leucine levels inhibit Sestrin-mediated mTORC1 suppression, potentially contributing to tumour progression. While Sestrin upregulation under stress conditions may counterbalance leucine-mediated inhibition, the functional impact of high leucine availability in cancers with low Sestrin expression remains unclear. Further studies targeting leucine transporters or modulating tumour amino acid metabolism could reveal additional strategies to restore Sestrin tumour-suppressive functions [126].

Overall, modulation of Sestrin activity through small molecules or metabolic interventions presents multiple avenues for enhancing cancer therapies (Figure 5). While direct Sestrin activation may reinforce tumour-suppressive mechanisms via mTORC1 inhibition, strategies aiming to inhibit Sestrins must carefully consider the context, due to the risk of inadvertently supporting tumour progression. Further research is required to define the specific contexts in which pharmacological or metabolic modulation of Sestrins may yield therapeutic benefit.

## 4. Future Directions

Despite substantial progress, the role of Sestrins in cancer remains complex and, at times, contradictory. To resolve these discrepancies and effectively leverage Sestrins in oncological contexts, several key areas require further investigation.

First, it is critical to elucidate the molecular pathways underlying Sestrin function in cancer. This includes confirming the regulation of STAT3 by Sestrins across various tumour contexts, defining the transcriptional interplay between Sestrins and Nrf2 in oncogenic settings, and clarifying the mechanisms by which Sestrins interact with mTORC2 and activate AKT signalling in cancer. Given the established regulation of Sestrins by leucine, future studies should determine whether additional Sestrin-mediated signalling pathways are similarly modulated by leucine availability.

Second, the relationship between Sestrins and metabolic stress requires deeper exploration. Pathways such as glutamine metabolism, amino acid metabolism, and ROS management are highly relevant, especially considering parallels between Sestrin function in metabolic diseases such as diabetes and in cancer. Given the complexity of cancer metabolism, this area represents a substantial and underexplored field for Sestrin research. The development of Sestrin–GATOR modulators, currently under investigation for neurological disorders, presents an exciting therapeutic avenue. Rational design of leucine analogues and leucine antagonists, based on structural insights into the Sestrin leucine-binding pocket, should be extended to cancer studies. In parallel, targeting deregulated leucine metabolism in tumours could enhance the therapeutic efficacy of Sestrin modulation.

Third, a comprehensive understanding of Sestrin post-translational modifications may reveal novel opportunities for therapeutic intervention, particularly in tumour contexts where such modifications may impair Sestrin function.

Finally, beyond their direct involvement in carcinogenesis, Sestrins contribute to cancer prevention by maintaining tissue integrity, as evidenced by their role in protecting the colon epithelium from inflammatory damage. Given the strong association between ageing and cancer incidence, the anti-ageing functions of Sestrins highlight their potential role in cancer prevention [41].

## 5. Conclusions

Sestrins are ubiquitously expressed proteins that are markedly upregulated in response to stress, including DNA damage and metabolic stress, both of which are frequently observed in transformed malignant cells. In response to a diverse array of transcription factors, Sestrins are on the frontline in the fight against cancer, but their implication in tumour initiation is limited. Evidence suggests that although Sestrins do not play a direct role in tumour initiation, they counteract several cancer-promoting events and are often impaired by oncogenic disruptions, such as the loss of p53 function. Studies consistently show that a reduction in Sestrin levels is highly correlated with tumour growth, impaired apoptosis, and poor prognosis across multiple tumour types. However, in certain contexts or metabolic conditions, increased Sestrin expression may instead support tumour progression.

Although the involvement of Sestrins in mTORC1 signalling is well established, their relationship with mTORC2 and survival factors like STAT3 remains poorly understood and warrants further study. Exploring these pathways further may provide insight into why Sestrins, despite their usual tumour-suppressive role, can sometimes facilitate cancer progression. Moreover, clarifying the molecular mechanisms of Sestrins could enhance their potential as prognostic and therapeutic markers, informing more precise treatment strategies.

Potentially targeting metabolic pathways, such as glutamine metabolism, is beneficial for tumours that exploit high Sestrin levels to protect themselves from metabolic stress, while supporting Sestrin expression in low-Sestrin tumours could enhance the effectiveness of conventional DNA-targeting therapeutic drugs, such as cisplatin or etoposide. In tumours with lower Sestrin expression, including those with p53 deficiency, Sestrins could be activated via alternative transcription factors.

The role of Sestrins’ leucine binding properties is currently poorly studied in cancers. Elevated leucine levels, common in many cancers, may inhibit Sestrin-mediated mTORC1 suppression, thereby promoting tumour growth. Therefore, future therapies should carefully integrate Sestrin expression levels, leucine metabolism, and broader cancer metabolic profiles to effectively exploit Sestrins as therapeutic targets.

## Figures and Tables

**Figure 1 cancers-17-01578-f001:**
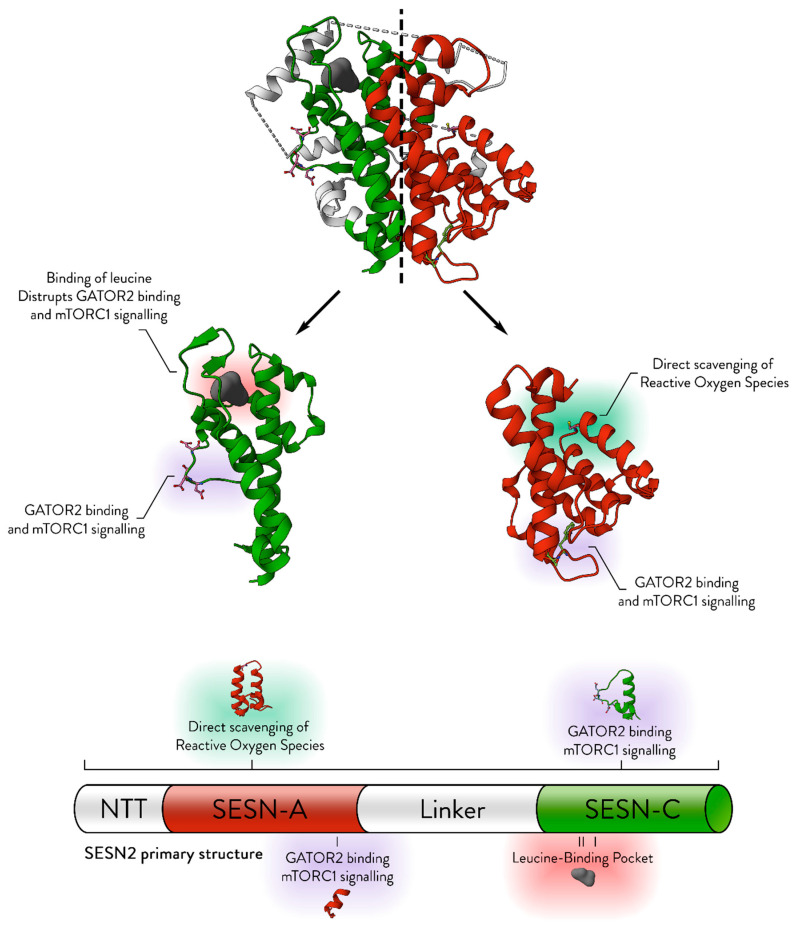
The structural basis for the established molecular roles of Sestrin. The structure of SESN2 (PDB ID: 5DJ4) is displayed, with its symmetrical nature emphasised by a central dashed line. The SESN-A and SESN-C domains, without the linker domain, are shown separately for detailed examination. Key structural features essential for the molecular functions of Sestrins are annotated. NTT refers to the N-terminal tail. The colour scheme highlights different domains: red for SESN-A, green for SESN-C, and white for the linker region. Modified from [2].

**Figure 2 cancers-17-01578-f002:**
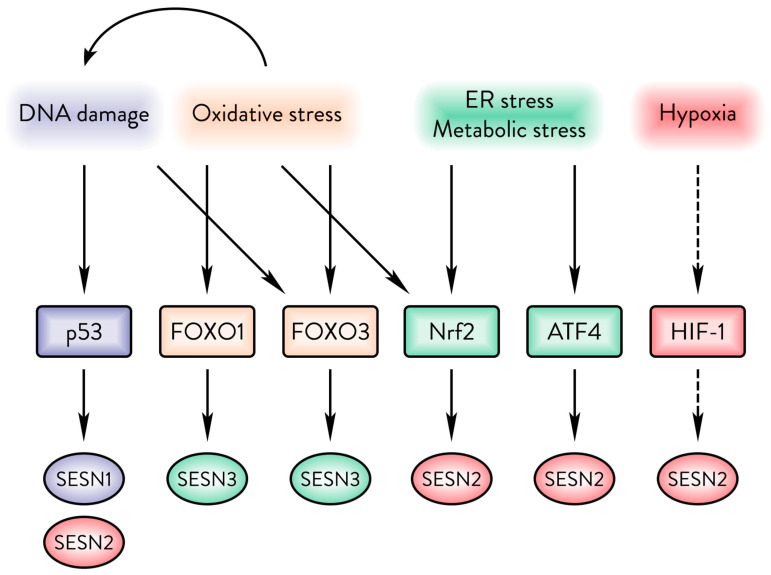
The Sestrin family responds to a variety of cellular stresses. DNA damage upregulates SESN1&2. Oxidative damage upregulates SESN3 but can also cause upregulation of SESN1&2 via p53. ER and energy stresses can upregulate SESN2 via Nrf2 or ATF4. Hypoxia upregulates SESN2, but it is unclear whether that occurs in a HIF-1α-dependent manner.

**Figure 3 cancers-17-01578-f003:**
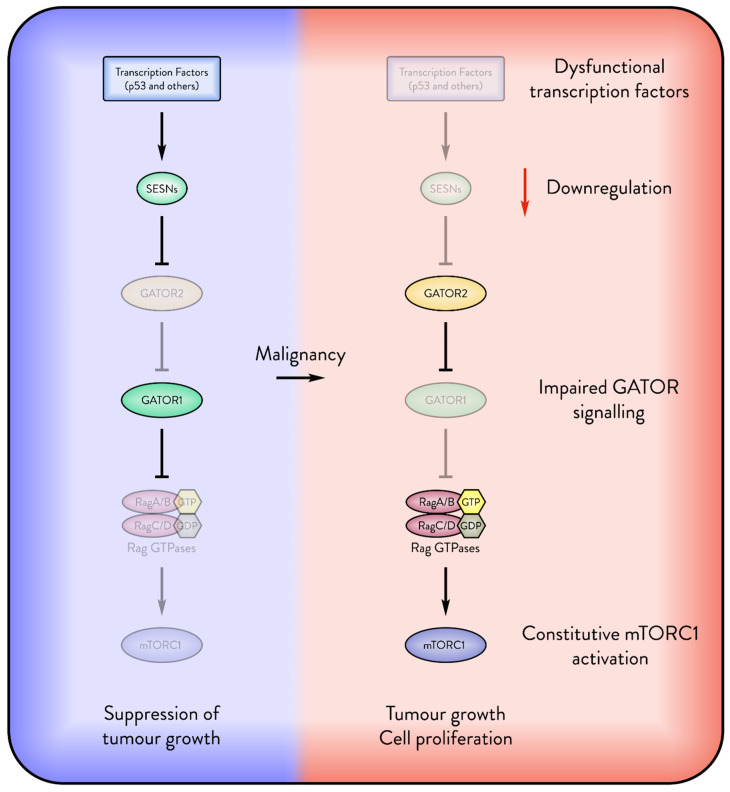
Cancers attempt to suppress the Sestrin inhibition of mTORC1. The figure displays several possible mechanisms by which malignant cells override the inhibition of mTORC1 by Sestrins (SESNs). The absence of transcription factors such as p53 downregulates the expression of Sestrins. Dysfunctional GATOR signalling may override Sestrin inhibition of mTORC1, enabling the cancer to freely express Sestrin for its pro-survival functions.

**Figure 4 cancers-17-01578-f004:**
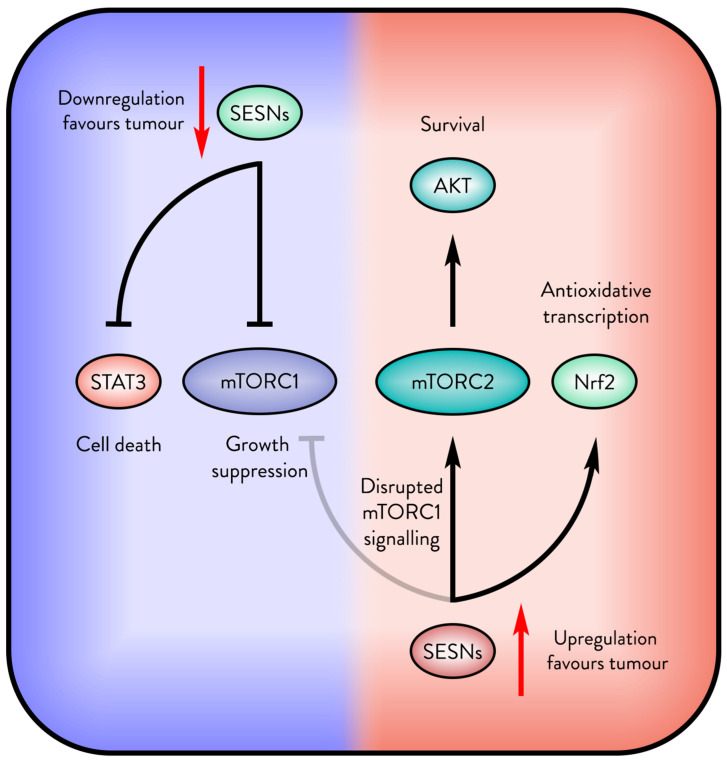
The opposite outcomes of Sestrin in cancer. Sestrins (SESNs) support tumour suppression, but when mTORC1 signalling is disrupted, Sestrins may be upregulated by the tumour for their antioxidative and survival signalling.

**Figure 5 cancers-17-01578-f005:**
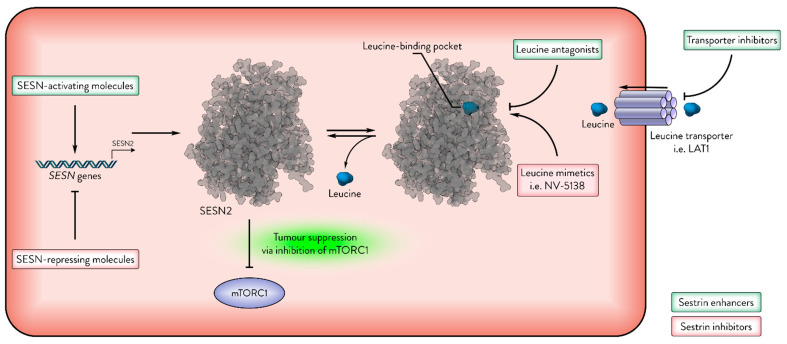
Sestrins as therapeutic targets in cancer. The activity of Sestrins (e.g., SESN2) can be modulated by leucine and other molecules. This figure illustrates several mechanisms of enhancing or inhibiting Sestrin activity. Green indicates Sestrin-enhancing molecules; red indicates Sestrin-inhibiting molecules.

**Table 1 cancers-17-01578-t001:** Expression and the biological role of Sestrins across various tumour types. The table summarises the expression patterns of Sestrin proteins in different cancers, the expression differences vs. normal tissues or modes of genetic manipulation, the impact on tumour progression (e.g., suppression or support), and the associated molecular mechanisms. Abbreviations: FL, follicular lymphoma; ↓, downregulation; ↑, upregulation; NSCLC, non-small cell lung cancer; TCGA, The Cancer Genome Atlas; HSPC, hormone-sensitive prostate cancer; HRPC, hormone-refractory prostate cancers; CRC, colorectal cancer; EMT, Epithelial-Mesenchymal Transition; N/A, not available.

Tumour Type	Model	Sestrin Expression	Genetic Manipulation	SESN’s Role	Mechanism/Effect	Reference
Follicular lymphoma	Primary tumours FL (n = 260)	SESN1 ↓	N/A	N/A	Reduced SESN1 expression by 6q21 locus deletion	[68]
	SU-DHL-10 cell line	N/A	SESN1 knockout	Suppress cancer	Inhibition of apoptosis in response to GSK126—EZH2 methyltransferase inhibitor	
Lung cancer	Cancer Profiling Expression Array (Clontech)	SESN1 ↓ SESN2 ↓	N/A	N/A	N/A	[30]
	Mice (n = 16)	N/A	Sesn2 knockout	Support cancer	Increased AKT phosphorylation; Decreased tumour size	
	A549 cell line	N/A	SESN1; SESN2; SESN1&2 knockout	Suppress cancer	Increased cell proliferation; Resistance to cell death by glucose starvation	
		N/A	siSESN2	Suppress cancer	Increased proliferation of cell line; Higher intracellular ROS	[43]
	A549; H460 cell lines	N/A	shSESN2	Suppress cancer	Resistance to death receptor-induced apoptosis; Regulation of XIAP protein expression	[66]
	A549; H460 cell lines	N/A	SESN1; SESN2; SESN1&2 knockout	Suppress cancer	Increased STAT3 phosphorylation; Increased cell proliferation; Resistance to apoptosis in response to genotoxic drugs	[71]
	Primary tumours NSCLC (n = 12)	SESN2 ↓	N/A	N/A	N/A	[33]
Liver cancer	TCGA liver cancer dataset (Low SESN3 n = 151, High SESN3 n = 214)	N/A	N/A	N/A	Higher SESN3 correlated with higher survival rates	[72]
	Mice (n = 10)	N/A	Sesn3 knockout	Suppress cancer	Increased tumour number; Increased tumour volume; Increased metastasis	
Prostate cancer	HSPC to HRPC progression	SESN3 ↑	N/A	Support cancer	Reduced expression of micro-RNA-708, which supresses SESN3 mRNA	[73,74]
Skin cancer	A431; A375; MEL624 cell lines	N/A	shSESN2	Support cancer	Decreased tumour volume; Decreased cell proliferation; Decreased AKT phosphorylation; Increased apoptosis in response to UVB and 5-fluorouracil	[31]
	Primary skin samples (n = 10)	SESN2 ↑	N/A	Support cancer	N/A	[31]
	Sebaceous skin carcinoma samples (n = 20)	SESN2 ↓	N/A	Suppress cancer	Loss of p53 correlated with absence of SESN2; Reduced SESN2 expression correlated with poor tumour differentiation	[77]
Colon cancer	Primary tumours CRC (n = 273)	SESN2 ↓	N/A	Suppress cancer	Low SESN2 expression predicted poorer survival	[57]
	HT-29; SW480; SW620; LoVo cell lines	SESN2 ↓	N/A	Suppress cancer	Immunofluorescence showed significant SESN2 downregulation in these cell lines	
	Mice (n = 11)	N/A	Sesn2 knockout	Suppress cancer	Increase in tumour number; Increase in tumour size; Increased mTORC1 activity; Increased sensitivity to colitis	[78]
	RKO cell line	N/A	shSESN2	N/A	Increased mTORC1 activity; Decreased AKT phosphorylation; Increased proliferation	
Endometrial cancer	Primary samples (n = 11)	SESN2 ↑	N/A	Support cancer	High mTORC1 activity despite SESN2 expression	[79]
	TCGA liver cancer dataset (Low SESN2 n = 74, High SESN2 n = 99)	N/A	N/A	N/A	Lower SESN2 correlated with higher survival rates	
	HEC-1A; Ishikawa cell lines	N/A	shSESN2	Suppress cancer	Increased mTORC1 activity; Increased cell proliferation; Increased intracellular ROS; Increased cell migration and EMT markers; Increased tumour volume and weight	
	Primary samples (n = 361)	SESN3 ↓	N/A	N/A	Possible repression of SESN3 via hyperactive methylation	[80]
Brain	Tet21N cell line (neuroblastoma)	SESN2 ↓	N/A	Suppress cancer	Demethylation by LSD1 represses SESN2 expression; Increased mTORC1 activity	[81]
	U87 cell line (glioblastoma)	N/A	siSESN2	Support cancer	Accumulation of PDGFRβ due to altered ubiquitination; Increased proliferation; Increased intracellular ROS	[82]
Cancer Stem Cell	Huh7 cell line	N/A	SESN3 knockout	Suppress cancer	Elevated stem cell markers Cd44, and Cd133; Interaction with Gli2 protein	[72]
	HCT-116; SW620 cell lines	N/A	SESN2 overexpression	Suppress cancer	Lower stem cell markers Sox2, Oct4 and CD44; Lower tumour volume and weight; Decreased expression of β-catenin and c-Myc	[94]

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
