# Peer review of "Sestrins in Carcinogenesis—The Firefighters That Sometimes Stoke the Fire"

_cancers, 2025, doi:10.3390/cancers17091578_

Round 1

Reviewer 1 Report

Comments and Suggestions for Authors

The current review, "Sestrins in carcinogenesis—the firefighters that sometimes stoke the fire, by Haidurov A et al., investigates the molecular mechanisms behind these seemingly contradictory effects on carcinogenesis and assesses the evidence for Sestrin's involvement across different cancer types.

However, the current version of the manuscript requires some revisions:

The authors discussed several cancer types and the known roles of sestrin proteins. It would be highly beneficial to include a table summarising the major findings across different cancers, along with the specific types of Sestrins involved.

While the authors have described the role of Sestrins in neuroblastomas, there is no mention of their involvement in glioblastomas (GBM). It would be helpful to clarify whether any relevant studies exist for GBM.

Sestrin proteins appear to be promising therapeutic targets; however, the manuscript does not mention any compounds or drugs that target these proteins. Including a brief paragraph on current challenges in drug development targeting Sestrins would greatly enhance the manuscript.

Author Response

We sincerely thank the reviewer for their careful evaluation of the manuscript, as well as for their valuable time and thoughtful feedback.

The authors discussed several cancer types and the known roles of sestrin proteins. It would be highly beneficial to include a table summarising the major findings across different cancers, along with the specific types of Sestrins involved.

We appreciate the reviewer’s insightful suggestion. We have organised the findings from the 'Sestrins in Cancer' section into a detailed Table 1, which is now included in the section and helps clarify the key points.

While the authors have described the role of Sestrins in neuroblastomas, there is no mention of their involvement in glioblastomas (GBM). It would be helpful to clarify whether any relevant studies exist for GBM.

We appreciate the reviewer’s important suggestion. We have reviewed the literature for studies on Sestrins in glioblastoma (GBM) and have now supplemented the manuscript with this cancer type.

Sestrin proteins appear to be promising therapeutic targets; however, the manuscript does not mention any compounds or drugs that target these proteins. Including a brief paragraph on current challenges in drug development targeting Sestrins would greatly enhance the manuscript.

We appreciate that the reviewer highlighted this point. Given the broad therapeutic potential of Sestrins, we have added a new section titled 'Sestrins as Therapeutic Targets in Cancer.' This section combines newly written material with concepts previously discussed in the Future Directions and Conclusions sections.

Reviewer 2 Report

Comments and Suggestions for Authors

The Haidurov and Budanov manuscript is a review article that provides a literature-based update on the role of sestrins in carcinogenesis. I agree with the authors that this is a good time to update the topic. The review perfectly covers the general pathways where sestrins play their sometimes contradictory roles. In addition, the authors also pay special attention to certain types of cancer. In my opinion, this part should be enchanted. In this case, I think that there is still a possibility to give more attention to comparisons of the roles of sestrins in different types of cancers, pointing out their potential as diagnostic markers and therapeutic targets for each particular case. In addition, in my opinion, the conclusions could also be more concrete, suggesting the directions of further research for specific areas, instead of the very general conclusions presented in the current version.  

Minor

- The simple summary does not mention that the review also covers such practical things as the potential of sestrins as diagnostic markers and therapeutic targets;

- More illustrations could be included.

Author Response

We sincerely thank the reviewer for their careful evaluation of the manuscript, as well as for their valuable time and thoughtful feedback.

The Haidurov and Budanov manuscript is a review article that provides a literature-based update on the role of sestrins in carcinogenesis. I agree with the authors that this is a good time to update the topic. The review perfectly covers the general pathways where sestrins play their sometimes contradictory roles. In addition, the authors also pay special attention to certain types of cancer. In my opinion, this part should be enchanted. In this case, I think that there is still a possibility to give more attention to comparisons of the roles of sestrins in different types of cancers, pointing out their potential as diagnostic markers and therapeutic targets for each particular case.

We appreciate the reviewer’s thoughtful comments regarding the 'Sestrins in Cancer' section. To enhance this part of the manuscript, we have added information on an additional cancer type (glioblastoma) and organised the findings into a new detailed Table 1, which summarises the roles and biological effects of Sestrins across different cancers.
Furthermore, given the broad and complex therapeutic potential of Sestrins, we have added a new section titled 'Sestrins as Therapeutic Targets in Cancer.' This section addresses the various perturbations affecting Sestrin function in tumours and discusses strategies to therapeutically modulate their activity, such as enhancing Sestrin expression.

In addition, in my opinion, the conclusions could also be more concrete, suggesting the directions of further research for specific areas, instead of the very general conclusions presented in the current version. 

We thank the reviewer for this valuable comment. We agree that the previous combined 'Conclusions and Future Directions' section was overly broad and mixed distinct ideas related to therapy, unresolved mechanisms, and future research priorities. To improve clarity, we have now divided it into two separate sections: 'Future Directions' and 'Conclusions.' These sections are more detailed, structured, and better aligned with their respective aims. We hope this revision provides the greater clarity requested by the reviewer.

Minor

- The simple summary does not mention that the review also covers such practical things as the potential of sestrins as diagnostic markers and therapeutic targets

We appreciate the reviewer’s observation. We have revised the Simple Summary to explicitly mention the potential of Sestrins as diagnostic markers and therapeutic targets.

- More illustrations could be included

We appreciate the reviewer’s suggestion regarding the inclusion of additional illustrations. In response, we have added a new figure (Figure 5. 'Sestrins as Therapeutic Targets in Cancer'), which highlights mechanisms that could be leveraged to therapeutically target Sestrins in cancer.

Round 2

Reviewer 2 Report

Comments and Suggestions for Authors

The manuscript has been improved and is now ready for publication.